# Quantifications of Mandibular Trabecular Bone Microstructure Using Cone Beam Computed Tomography for Age Estimation: A Preliminary Study

**DOI:** 10.3390/biology11101521

**Published:** 2022-10-18

**Authors:** Arshiya Tabassum, Mansharan Kaur Chainchel Singh, Norliza Ibrahim, Subramaniam Ramanarayanan, Mohd Yusmiaidil Putera Mohd Yusof

**Affiliations:** 1Center for Oral and Maxillofacial Diagnostics and Medicine Studies, Faculty of Dentistry, Universiti Teknologi MARA Selangor, Sungai Buloh 47000, Selangor, Malaysia; 2Institute of Pathology, Laboratory and Forensic Medicine (I-PPerForM), Universiti Teknologi MARA Selangor, Sungai Buloh 47000, Selangor, Malaysia; 3Department of Oral and Maxillofacial Clinical Sciences, Faculty of Dentistry, University of Malaya, Kuala Lumpur 50603, Federal Territory of Kuala Lumpur, Malaysia; 4Department of Public Health Dentistry, Indira Gandhi Institute of Dental Sciences, Nellikuzhi P.O., Kothamangalam, Kerala 686691, India; 5Department of Forensic Odontology, Faculty of Dental Medicine, Universitas Airlangga, Surabaya, Jawa Timur 60132, Indonesia

**Keywords:** trabecular bone microstructure, age estimation, cone beam CT

## Abstract

**Simple Summary:**

Studies pertaining to age-related changes in the human jaw are very limited. Age-related changes for forensic assessments were merely based on the qualitative architecture and external geometry. With advancements in imaging technology, the paradigm has shifted and more emphasis is laid on the internal trabecular architecture, as this imparts more complexity and strength when compared to other determinants. The desired objective of this study was to draw useful associations between trabecular microarchitecture and chronological age. In this study, we characterize trabecular bone microstructure in the human mandible in 20 subjects ranging from 22 to 43 years using retrospective cone beam computed tomography image datasets. The images were post-processed via a semi- automated threshold guided approach and reconstructed using AnalyzeDirect 14.0 software to correlate trabecular number (Tb. N), trabecular thickness (Tb. Th), trabecular separation (Tb. Sp), trabecular surface density (Tb- BS/TV) and trabecular bone volume (Tb- BV/TV) with the individual’s chronological age and sex. Statistically significant negative correlations were observed in trabecular number (r =−0.489) and trabecular surface density (r = −0.527) in relation to chronological age. The characterization and quantification of the trabecular bone microarchitecture can immensely serve as a digital imprint in chronological age estimation.

**Abstract:**

The aim of this study is two-fold: first, to correlate the values for each of the trabecular bone microstructure (TBM) parameters to the individual’s chronological age and sex, thereby facilitating the assessment of potential age and sex-related changes in trabecular bone microstructure parameters in the mandible; and second, to quantify the trabecular microstructural parameters in relation to chronological age. Twenty cone-beam computed tomographic (CBCT) scans were retrieved retrospectively from a database of adult patients with ages ranging in age from 22 to 43 years. In the mandible, the volume of interest included the inter-dental space between the second mandibular premolar and the first mandibular molar, as well as the trabecular space beneath and between the apices. Using the AnalyzeDirect 14.0 software, the DICOM images of CBCT scans were pre-processed, transformed, segmented using a novel semi-automatic threshold-guided method, and quantified. In addition, TBM parameters were derived, and statistical analysis was conducted using a Pearson correlation test with two tails. All parameters exhibited no statistically significant differences (*p* > 0.05) between chronological age and sex. Statistically significant negative correlations were found between Tb. N (r = −0.489), BS/TV (r = −0.527), and chronological age (*p* = 0.029 and *p* = 0.017, respectively). Only Tb. N and BS/TV exhibited an inverse relationship with chronological age. Numerous studies have quantified the trabecular architecture of the jaw bones, but none have found a correlation between the quantified trabecular parameters and chronological age. The digital imprints produced by radiographic imaging can serve as biological profiles for data collection.

## 1. Introduction

Modeling and remodeling of bone are the processes that dictate structural and functional integrity, while the latter is a frequent phenomenon [1]. It is estimated that approximately 10% of the human skeleton remodels each year [2]. Age estimation by qualitative methods is discontinuous and provides mean and age range estimates for each discrete phase of morphological change in the human skeleton and thereby subjected to large errors [3]. The term “bone quality” is not so simple to define [4]. There is no clear consensus on the definition of bone quality, but, in general, it encompasses multiple aspects of bone physiology, the degree of mineralization, the morphology, and the type of trabecular pattern [5]. It is thus referring to those structural, material, and cellular qualities of bone that regulate its mechanical competency at various length scales. The overall bone geometry (size and shape), as well as its microstructural (trabecular and cortical) and nanostructure arrangements, are characteristic of structural qualities (woven and osteonal bone) [6,7,8]. However, “bone quantity” on the other hand is easily defined as the amount of bone height and the width of the alveolar crest at an edentulous site [5]. The quantitative methods of age estimation require destructive sampling of bone [9], but are far superior to qualitative methods which solely rely on bone geometry and the architectural makeup of the bone [10]. Assessment of trabecular architecture from plain film radiographs also depends on surface characteristics and is mainly qualitative [9]. Previous studies on age-related changes in the jaw bones were purely qualitative and were based solely on macro-structural parameters (mandibular dimensions, morphometry) such as mandibular ramus height, bigonial width, and mandibular angle [11]. Many studies have also focused on bone quality assessments for implants but yielded controversial results [12]. Bone with poor cortical density and large marrow spaces leads to poor osteo-integration but even dense cortical bone does not yield favorable results [12,13]. The whole implant interface is in maximum contact with the trabecular microarchitecture, but previous studies concentrated only on bone mineral density [14], which was in fact in the past considered the most important factor for assessing bone quality and strength, but lately most studies have laid focused on the trabecular microstructure parameters of the bone [15]. However, quantification of the trabecular microstructure demanded excellent spatial resolution of the imaging modalities [16]. The examination, assessment, and quantification of jaw architecture by CBCT would be the most advantageous outcome as it provides a digital imprint to be revisited, re-tested, and re-evaluated by future researchers, thereby avoiding destructive bone sampling at the solely quantitative assessment site and objective based on the microarchitectural trabecular bone remodeling [9]. This study was prompted by findings in the trabecular microstructural parameters that were extensively studied in the axial skeleton [9,17], (femoral, vertebrae, tibia, and radius) but was mostly done on cadaveric specimens correlating with chronological age using various imaging modalities [17].

Subsequently, though the mandible served as an important bone for forensic investigation, age-related quantification of the trabecular bone microstructure (TBM) in the human jaws was not attempted. Thus, there is no data in the current literature citing the quantification and correlation of TBM parameters with chronological age using advanced imaging modalities. It is also assumed that various factors such as age, sex, and status of dentition have an influence on the TBM in the mandible [18]. This is particularly important in cases of skeletal remain identification such as homicide, and on underage persons of unknown age whereby the age assessment is of crucial. 

To date, qualitative bone assessments based on surface characteristics and macro-structural parameters of the jaw bones are relied upon for age estimation to study age-related changes [9]. Moreover, as the age progresses, the qualitative parameter such as the mandibular trabecular volumetric bone mineral density (vBMD) did not synchronously reflect with the spinal vBMD. The possible reasons for this could be that the mandible sustains additional mechanical stresses from occlusion, its complex process of ossification, and its distinct embryonic origin, thereby downplaying the qualitative parameters for studying age-related changes [19]. Hence, the need for an objective-based quantitative assessment tool coupled with a non-invasive imaging modality would prove noteworthy for studying age-related changes in the trabecular microstructure of the mandible.

With the advancement in 3D imaging, quantification by non-invasive means is possible, whereby histomorphometric assessments demand invasive methods, though histomorphometry was considered the gold standard [20]. However, after the emergence of micro-computed tomography (µCT), and the quantification of trabecular architecture by µCT yielded promising results, and its ability to attain resolutions up to 0.5 µm, it is currently tagged as a non-destructive gold standard [21]. Previous studies have validated CBCT to be used as an assessment tool to quantify trabecular microarchitecture [22,23,24,25,26]. The studies suggested that morphometric assessments by most CBCT machines and high-resolution multi-scanner computed tomography (MSCT) are reliable, though they over-estimated the trabecular microarchitecture the alveolar network remained the same between them and the reference gold standard (µCT) [27].

Consequently, the current study aimed to assess each of the TBM parameters of the mandible (Trabecular Number (Tb. N), Trabecular thickness (Tb. Th), Trabecular separation (Tb. Sp), Trabecular bone volume (Tb- BV/TV %), and Trabecular surface density (Tb- BS/TV mm^2^/mm^3^) and correlate them with the individual’s chronological age and sex using retrospective CBCT dataset. Thus, this study is the first attempt ever to use CBCT images in quantifying the TBM parameters and further developing a prediction model to estimate the chronological age to aid in forensic applications.

## 2. Materials and Methods

This pilot study was a cross-sectional observational study for secondary data analysis. The retrospective CBCT datasets in DICOM (Digital Imaging and communications in medicine) that met our protocol for voxel size (90 µm), tube voltage (80 kV), tube current (3 mA), and field of view (50 × 50 mm) were obtained following the ethical approval from the institutional review board. The CBCT datasets of fully dentate patients which were evaluated for impactions, pre-orthodontic treatment, and TMJ disorders were considered for the study. Additionally, the CBCT datasets of patients without any previous history of systemic diseases or that alter bone metabolism/previous history of trauma, radiation and chemotherapy, and calcium/bisphosphonate therapy were included in this study. Further, all patients with potential generalized systemic, metabolic, and nutritional illnesses of bone, those having received orthodontic treatment or any kind of therapies for bone disorders/non-bone disorders that can alter the composition and density of bone, and subjects with a previous history of periodontal treatment in the vicinity of the region of interest were excluded from the study. Secondly, the bone quality can change according to secondary causes such as high alcohol consumption and smoking; thus, the records of such patients were also excluded from the study.

The study was conducted in five steps (Selection of Region of interest (ROI), Data Preprocessing, Segmentation, Bone microstructure assessment (BMA), and Intra-observer agreement). Analyze 14.0 software (Analyze Direct, Overland Park, KS, USA) with a BMA add-on was employed in this study for estimating the TBM parameters. It is a pre-clinical research software utilized for the evaluation and quantification of 3-D image data. The BMA add-on provides excellent user-guided isolation of bone from non-bone tissue by manual, semiautomatic, and automatic threshold-guided segmentation of cortical and trabecular components of bone. It utilizes the segmented object map to automatically calculate the quantification of TBM parameters. 

### 2.1. Selection of Region of Interest (ROI)

Based on previous literature, the mandible distal to the mental foramen and the region posterior to the second premolar were regarded as the standard sites for evaluating age-related alterations in vivo [28] because it exhibited the fewest intra- and inter-individual variability in anatomic form, size, structure, and function (Figure 1). 

Morant et al. also opined that sampling is done with reference to the mental foramen because it is a highly stable structure with excellent demarcation between alveolar and basal mandibular bone [29]. As a result, our ROI is likewise located in the inter-radicular region between the second mandibular premolar and first mandibular molar (about 6 mm away from the mental foramen) and 2.5–3.0 mm apical to the alveolar crest (based on the maximal range of effects of bone loss from periodontal disease) [30]. The volume of interest (VOI) to be quantified in the selected ROI is the trabecular architecture in the interdental area between the second mandibular premolar and first mandibular molar as well as the trabecular area beneath and between the apices that are devoid of teeth, canals and foramen in the vicinity comprising of trabecular bone superiorly and until the basal bone inferiorly. The VOI was defined by the top slice (i.e., the slice where the second premolar disappears) and the bottom slice (i.e., the slice where the second molar appears). The dataset comprised an average of approximately 400 slices. The middle slice of the data set containing the desired VOI was chosen as the reference slice. Then, the VOI defined for analysis was set at 20 slices back and forth from the reference slice. Hence, a total of 40 slices were investigated. On each of the slices, the trabecular components were delineated by manual and semiautomatic segmentation (fill holes and region grow) by employing adaptive thresholding. Segmentation was automatically enabled by the software, which was segmented and analyzed in all the three planes (coronal, sagittal, and axial planes). The analysis intended was to quantify all of the trabecular bone defined within the VOI. 

### 2.2. Data Preprocessing

The preprocessing of data was a prerequisite to facilitate the bone sample to have isotropic voxels. The process tab in the software enables to apply spatial filters (medial filter) and kernel size 3 × 3 × 3 to enhance segmentation. After the processed volume was saved, it was later transformed using the transform tab which enables reorienting the data in the coronal (oblique plane) and sagittal planes (perpendicular axis). Later, the image was transformed by enabling the Apply Matrix tab. Further, the sub−regioning was done, which enabled to crop the image data pertaining to the volume of interest in all three orientations (Figure 2) to facilitate quick processing of data, thereby reducing the overall size of the dataset. The transformed image was stored in the workspace.

### 2.3. Segmentation (Manual and Semi-Automatic)

The segmentation process was carried out sequentially by segmenting the bone into the cortex and trabecular components and utilizing an adaptive thresholding algorithm implementing the threshold tab (Figure 3). 

The segmentation was done both manually (tracing) and with the semiautomatic modes (using the region grow and fill holes) option to segment the under-segmented areas within the trabecular architecture by advancing the slice-by-slice option in all orthogonal planes. Further, the trabecular component was segmented into trabecular septae, cortical bone and inter-trabecular space by adaptive thresholding (Figure 4). The final object map was color−coded as per the software requirements to be loaded into the BMA- add-on and saved in the desired folder.

### 2.4. Measurement and Tabulations via BMA

The saved final object map was loaded into the BMA add−on, and later quantification and assessment of the trabecular parameters were exported directly into the destined folder by opting the measure bone. The measurements of desired trabecular parameters (BV/TV, BS/TV, Tb. N, Tb. Th, Tb. Sp) were generated (Figure 5). 

The trabecular volume fraction (BV/TV) is defined as the ratio of segmented trabecular bone volume to the total volume of the trabecular volume of interest. It is measured in (%) units. The ratio of the segmented trabecular bone surface to the total trabecular volume of the volume of interest is known as trabecular surface density (BS/TV) and is measured in the units of mm^3^/mm^2^. The trabecular number (Tb. N) is the average number of trabeculae per millimeter and is measured as 1/mm. Trabecular thickness (Tb. Th) refers to the mean thickness of the trabeculae, while trabecular separation (Tb. Sp) relates to the mean distance between trabeculae. Both Tb. Th and Tb. Sp are measured in millimeters (mm). All *p*-values are two-tailed. Statistical significant was set at 0.05, and analyses were conducted using RStudio version 0.97.551—© 2009–2012 RStudio, Inc. software (Boston, MA, USA).

### 2.5. Intraobserver Reliability

All the 20 CBCT datasets were selected to analyze intra-observer reliability. Intra-observer reliability was estimated between the measurements performed two weeks apart. For intra-observer reliability and to reassess CBCT images, the intra-observer underwent rigorous training/calibration for certain trabecular parameters pertaining to segmentation and thresholding. Later, intraclass coefficient (ICC) statistics were applied to check the reliability before the commencement of the main study.

## 3. Results

A total of 20 retrospect CBCT DICOM images were analyzed for various parameters. Of the 20 datasets, 55% (*n* = 11) of the records belonged to the male and the rest 45% (*n* = 9) were of the female. The mean age of patients included in the study was 26.6 ± 5.9 years (with a range of 22–43 years) (Table 1). All patients came from the same geographic and ethnic origin. 

Further, a reliability analysis was undertaken using Intra-Class Coefficient (ICC). A two-way mixed model was used, as the raters were the same for each item. Table 2 outlines both single measure ICC and average ICC. Concerning the single measure ICC, it is observed that there was moderate intra-observer reliability with respect to TV, BS, Tb. N, and the rest of the parameters had poor intra-observer reliability.

The aimed parameters were statistically assessed using a two-tailed Pearson correlation test. The mean trabecular volume (TV) was 1647.85 ± 934.78, bone volume (BV) was 741.65 ± 544.19 and bone surface (BS) was 2653.45 ± 1453.22. Regarding the structural parameters of the bone, the mean bone volume fraction (BV/TV) was 44.40 ± 14.77, mean specific surface was 1.75 ± 0.64, and mean bone surface density (BS/BV) was 4.00 ± 1.21. The trabecular number (Tb. N), trabecular thickness (Tb. Th), and trabecular separation (Tb. Sp) were 0.443 ± 0.15, 1.25 ± 0.55, and 2.05 ± 0.75, respectively (Table 3).

When the correlation of the parameters with age and sex was analyzed, a significant negative correlation existed between BS/TV and Tb. N in relation to age, respectively (r = −0.527; *p* = 0.017 and r = −0.489; *p* = 0.029). Consequently, none of the other parameters showed a significant correlation with age or sex (Table 4).

## 4. Discussion

The ultimate objective of this field of study was to identify useful associations between trabecular bone microarchitecture and chronological age. The uniqueness of each individual in the human race is limited to one’s face and hands. As the bones and teeth of the craniofacial skeleton display excellent preservation over time, these structures are hence most commonly used for identification [31].

Todd’s radiographic study was pioneering research of the pubic bone that demonstrated an intrinsic relationship between age and the degeneration of the trabecular network [32]. The techniques employed in this study address two major issues that plague many existing age estimation analyses: (1) non-destructive sampling and (2) quantitative assessment.

Based on the previous anthropological studies, the segment of the mandible distal to the mental foramen and posterior to the second premolar was considered the typical site for evaluating age-related alterations in vivo [28] because it exhibited the least intra- and inter-individual discrepancies in anatomic shape, size, structure, and function. As a result, our ROI was also centered here on the inter-radicular area between the second mandibular premolar and first mandibular molar (roughly about 6 mm away from the mental foramen) and 2.5–3.0 mm apical to the alveolar crest (based on the widest possible range of effects of bone loss from periodontal disease) [30].

It has been noted that trabecular patterns vary across age and sex [33]. The present study had a statistically significant negative correlation between Tb. N and age. This study’s findings were in line with those of a prior study, which showed that the relationship between each of these measures and age was negative and that trabecular density (Tb. N) decreased with increasing age [9]. Further, we correlated Tb. N, and the sex values found in our study were similar to those reported by published literature, where there was no significant correlation established [34]. Another study on vertebral bodies and iliac crest revealed that the Tb. N decreased significantly with age, both in females and males [35,36]. The majority of these studies only included the most basic morphometric indices, such as bone volume fraction (BV/TV), trabecular thickness (Tb. Th), trabecular separation (Tb. Sp), and trabecular number (Tb. N), and thus contributed scant information on the general trabecular structure of the bone. The significance of BS/TV (Trabecular surface density) was first highlighted by Dessel et al. in the year 2017 [27]. Hence, we have included this parameter in our study, which showed a negative correlation with respect to chronological age inferring reduced trabecular surface density as age progresses. To the best of our knowledge, there are no available data on the correlation between BS/TV and age in the published literature on the human jaws.

Furthermore, none of the other parameters in our study correlated with age and sex. The reason for this could be due to the small sample size. It can also be attributed to the fact that different techniques and measuring units were used in previous studies by different researchers in estimating and expressing TBM assessments; hence, a definite comparison could not be established. Moreover, a scholarly literature emphasized and concluded that histomorphometry, micro-CT, and other various modalities provide complementary information regarding jawbone microarchitecture, but the poor agreement between the methods warns that their results should not be used interchangeably [37].

The intra-observer reproducibility of trabecular bone microstructure measurements was moderate. At any rate, the measurements were repeated twice, and the ICC results showed moderate intra-observer reliability (ICC ≥ 0.56) between the first and the second measurement. Therefore, the average of the two measurements was calculated for further analysis. Previous studies mainly reported excellent reliability for CBCT scan parameters on trabecular bone microstructure measurements [27,38]. This discrimination of reliability between our study is mainly due to the disparity in voxel sizes and the field of view used. These earlier studies employed voxel sizes in the range of 80 to 125 µm^3^ in contrast to the 250 µm^3^ employed in the current study. Further images in previous studies were binarized, followed by automatic segmentation, whereas our study employed semi-automatic coupled with manual segmentation with no prior binarization of the images. Moreover, most of the previous studies were undertaken in the cadaveric specimens which distinguish the current study where we analyzed the CBCT images from the live specimens. This intraobserver variability can be significantly larger in complex tasks, such as three-dimensional outlining, than in other medical situations where the outcome is a diagnosis or rating. The variability is an inherent function of the unique radiotherapy outlining clinical paradigm and does not constitute a problem [39].

From the previously published literature, it was noted that the Tb. N, connectivity density, bone volume, and mineral density decreased with age. There was an increase in Tb. Sp with increasing age [40]. The normative tables between age and BV/TV are usable in anthropology, which may act as an alternative to determine the age at death [41]. Further, it is documented that as the age increases, the horizontal Tb. Th, and horizontal and vertical Tb. N significantly decreases irrespective of sex. Further extension of the study on vertebral bodies and iliac crest revealed the ratio of trabecular bone volume to total volume, connectivity density, and Tb. N decreased significantly with age, both in females and males [35,36]. Similar findings have been highlighted in other studies [42,43,44,45,46].

There are studies that further correlated age and sex-related changes in various bones. With advancing age, in women Tb. N declines, and Tb. Sp increases and BV/TV had age-related decreases in both women and men [45]. These findings signify the importance of studying the trabecular microstructure of human bone, which may help in the estimation of age.

This was the first study that tried to standardize the methodology by using the adaptive thresholding method in evaluating the trabecular microstructure of the mandible. Under thresholding or over thresholding has its effect on trabecular parameters. Compared to manual segmentation, auto-segmentation has many drawbacks, due to which a combination of manual and semiautomatic segmentation has been used in this study. In earlier published literature, it was suggested that when evaluating the structure of the jawbone, it is ideal to choose adaptive thresholding [46].

Thus, the age assessment of human skeletal remains is an important step in building a biological profile in both archaeological and forensic applications. The quantitative or objective evaluation of trabecular microstructural parameters of mandibular bone using CBCT can act as an obliging tool. A limitation of this study is that the VOI pertaining to the trabecular microarchitecture was assessed from the trabecular bone until the basal bone in its entirety, with no partitions made between the trabecular bone and the basal bone. In addition, the study requires age interval stratification on its data to determine the specific stage(s) of TBM with the individual’s chronological age.

## 5. Conclusions

The present study was accomplished using a non-invasive imaging modality to perform an objective-based quantitative assessment for studying age-related changes in the selected trabecular microstructure parameters of the mandible. The results of the correlation of chronological age with the trabecular microarchitectural parameters of 20 CBCT DICOM images indicate that high-resolution CBCT of the trabecular bone structure may be useful for quantifying differences in mandibular trabecular microarchitectures. Further, this study, in the future, will serve as a preliminary analysis to devise a prediction model for depicting the chronological age based on the TBM parameters. A large and stratified dataset on age interval is recommended for future study.

## Figures and Tables

**Figure 1 biology-11-01521-f001:**
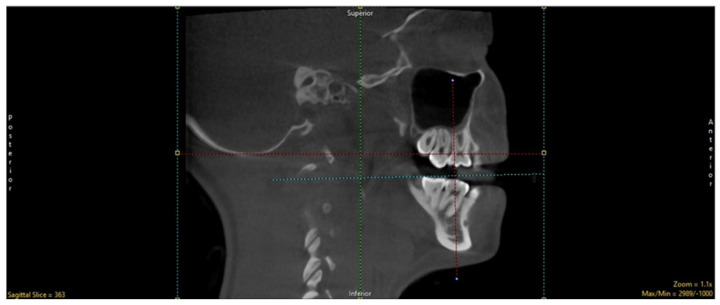
Selection of the region of interest (the dotted lines represent the orientation for the selected ROI).

**Figure 2 biology-11-01521-f002:**
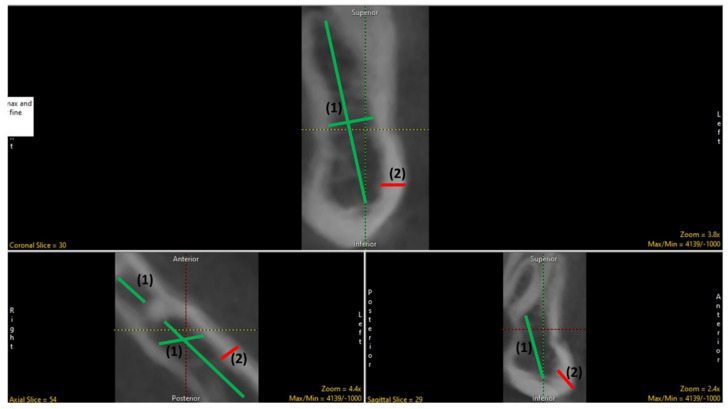
The sub-regioning of data. (1) trabecular bone; (2) cortical bone.

**Figure 3 biology-11-01521-f003:**
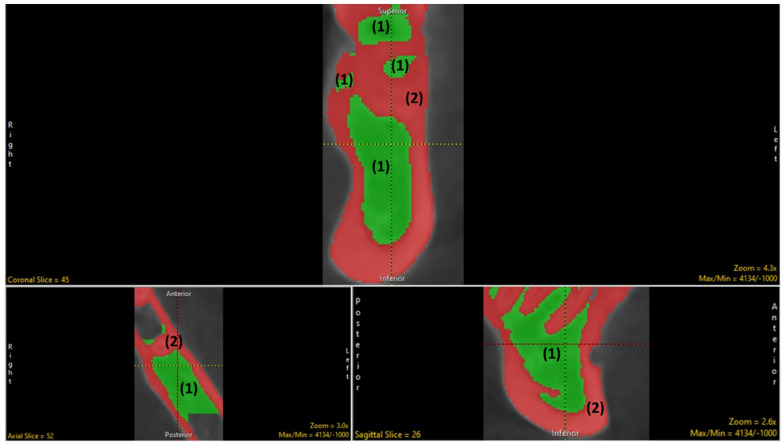
Bone segmentation into the (1) trabecular bone; (2) cortical bone.

**Figure 4 biology-11-01521-f004:**
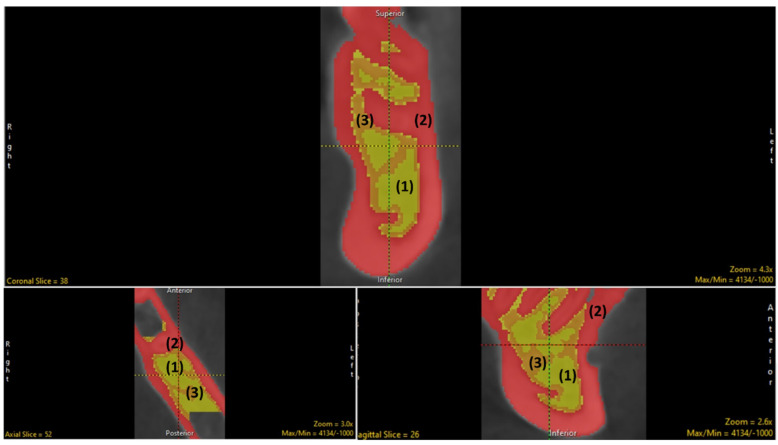
The trabecular component segmented into (1) inter-trabecular space, (2) cortical bone and (3) trabecular septae.

**Figure 5 biology-11-01521-f005:**
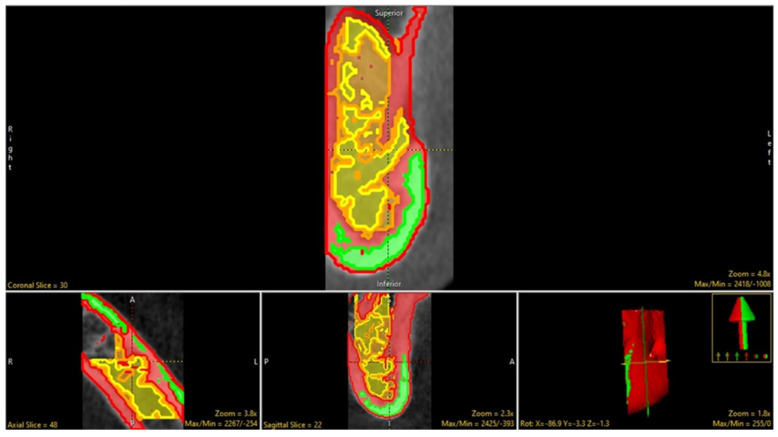
The measurement of desired trabecular parameters via BMA (green, yellow, and red denote inter−trabecular space, cortical bone and trabecular septa, respectively; A: Anterior; P: Posterior; L: Left; R: Right).

**Table 1 biology-11-01521-t001:** A profile of the study subjects.

Sex	N (%)
Male	11 (55%)
Female	9 (45%)
**Age**	
Mean ± SD	26.6 ± 5.9
Range	22–43 years

**Table 2 biology-11-01521-t002:** The intra-observer reliability.

Variable	Single Measure ICC	Average Measure ICC
TV	0.567	0.724
BV	0.318	0.482
BS	0.411	0.583
Tb. N	0.386	0.557
Tb. Th	0.052	0.100
Tb. Sp	0.092	0.169
Tb. Th. SD	0.037	0.071
Tb. Sp. SD	0.434	0.606

**Table 3 biology-11-01521-t003:** The mean and standard deviation (SD) of the TBM parameters.

Parameter	Mean ± SD
TV	1647.85 ± 934.78
BV	741.65 ± 544.19
BS	2653.45 ± 1453.22
BV/TV	44.40 ± 14.77
BS/TV	1.75 ± 0.64
BS/BV	4.00 ± 1.21
Tb. N	0.443 ± 0.15
Tb. Th	1.25 ± 0.55
Tb. Sp	2.05 ± 0.75
Tb. Th. SD	0.35 ± 0.49
Tb. Sp. SD	0.90 ± 0.45

**Table 4 biology-11-01521-t004:** The correlation between age and sex with the TBM parameters.

Parameter	Age		Sex	
	r Value	*p* Value	r Value	*p* Value
TV	0.076	0.749	−0.156	0.512
BV	0.051	0.832	−0.0258	0.272
BS	−0.079	0.740	−0.260	0.267
BV/TV	−0.152	0.523	−0.402	0.079
BS/TV	−0.527	0.017 *	−0.283	0.227
BS/BV	−0.219	0.354	0.255	0.278
Tb. N	−0.489	0.029 *	−0.379	0.099
Tb. Th	0.081	0.736	−0.234	0.320
Tb. Sp	0.016	0.946	0.075	0.754
Tb. Th. SD	−0.167	0.483	−0.032	0.895
Tb. Sp. SD	0.004	0.987	−0.023	0.923

* statistically significant (*p* < 0.05)

## Data Availability

The datasets generated and analyzed during the current study are not publicly available due to the security of data but are available from the corresponding author on reasonable request via email.

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
