# Peer review of "Quantifications of Mandibular Trabecular Bone Microstructure Using Cone Beam Computed Tomography for Age Estimation: A Preliminary Study"

_biology, 2022, doi:10.3390/biology11101521_

Round 1

Reviewer 1 Report (Previous Reviewer 3)

This article aimed to establish the relationship between bone microstructure (TBM) parameters and individual's chronological age and gender. However, the related research has been conducted by other resaerchers:

https://doi.org/10.5115/acb.20.269

https://doi.org/10.1016/j.bone.2021.116094

https://doi.org/10.1097/SCS.0000000000005069

To explore the correlation between TBM and individual’s information at the statistical level, 20 samples were too small. In addition, the authors should use quantitative analysis instead of qualitative analysis. Can the author give a calculation method based on a large amount of data?

Author Response

We thank the reviewers for the remarks.

The three studies that have been cited by the reviewer are not similar with our work. The first two studies were studying about the cervical and lumbar bone using micro CT. Our study utilized cone beam CT (CBCT) and we did not use micro CT. The last cited article was utilizing CBCT and the study was conducted to detect foramen of Huschke which was not our objective.

Our study utilized a small sample size due to the fact the conducted research was exploratory and a pilot study. For better clarity, we have amended the title to include 'a pilot study'.

Reviewer 2 Report (New Reviewer)

The idea of this article is interesting and worth considering for a forensic audience. The authors have done valuable work in analyzing CBCT scans and have adequately described the method. However, the results are not as convincing as they should be. The manuscript is worth publishing, but unfortunately not in its current form. Please consider my recommendations below.

1. first, I suggest the authors to use "preliminary study" in the title of the manuscript.

2. an important point to consider is the size of the sample studied. I realize that much detailed work needs to be done to evaluate each scan. However, if one wants to study the relationship between age and trabecular bone microstructure, the sample studied should be larger to draw such a conclusion, or age categories such as 20-30, 30-40, and over 40 years could provide more comprehensive conclusions. At the very least, this should be discussed as a possible limiting factor of the study.

3. in Figure 4 - number 2 in the figure should be described/referred to as cortical bone in the title of the figure.

4. Figure 5 - all "color parts" should be named/described in the title.

5. in the Material and Method section, I would at least include the age range and percentage of subjects in relation to gender.

6. the names of the figures and tables should be carefully revised; they do not have to start with "this is".

7.Table 2 refers to descriptive statistics only, not Pearson correlation results.

8.All abbreviations used in the tables should be explained below the table, as each table should be independently readable and understandable.

9.Please indicate the significance level, if it is 0.05, then in table 3 two parameters show significant correlation with age (BS /TV and Tb. N).

10. I suggest performing an independent-samples t-test or its nonparametric version (Mann-Whitney) to determine sex differences in all parameters. Performing a partial correlation (evaluating the control variable) might yield interesting results, please reconsider. I would also run an age correlation with respect to gender.

11. in the results section, I would start with the ICC since the other results depend on it.

12.The conclusion is written too presumptuously given the insignificant correlation analysis or not so strong correlation (considering the small sample size) and the poor ICC. According to the present result, I think that an age range whose average age is closer to the minimum age of the sample could bias the results.

Author Response

We would like to thank the reviewer for the remarks.

1. first, I suggest the authors to use "preliminary study" in the title of the manuscript.

Response: We agree with this suggestion and we have amended the title to include the preliminary study.

2. an important point to consider is the size of the sample studied. I realize that much detailed work needs to be done to evaluate each scan. However, if one wants to study the relationship between age and trabecular bone microstructure, the sample studied should be larger to draw such a conclusion, or age categories such as 20-30, 30-40, and over 40 years could provide more comprehensive conclusions. At the very least, this should be discussed as a possible limiting factor of the study.

Response: We have added discussions pertinent to this suggestion in page 11.

3. in Figure 4 - number 2 in the figure should be described/referred to as cortical bone in the title of the figure.

Response: Thank you for the suggestion. We have amended the title of Figure 4 accordingly.

4. Figure 5 - all "color parts" should be named/described in the title.

Response: We have included the description of the color accordingly.

5. in the Material and Method section, I would at least include the age range and percentage of subjects in relation to gender.

Response: The percentage of the gender has been reported in the Results section as follows.

A total of 20 CBCT DICOM images were analyzed for various parameters.  Of the 20 datasets, 55% (n=11) of the records belonged to the male gender and the rest 45% (n=9) were of the female gender. 

6. the names of the figures and tables should be carefully revised; they do not have to start with "this is".

Response: Thank you for this remark. We have corrected the title for figures and tables accordingly. 

7. Table 2 refers to descriptive statistics only, not Pearson correlation results.

Response: Thank you for pointing this out. Table 2 is indeed referring to descriptive statistics of mean and standard deviation. We have corrected the tile accordingly.

8. All abbreviations used in the tables should be explained below the table, as each table should be independently readable and understandable.

Response: Thank you for this remark. Due to the abundance of abbreviations used in the tables, we have made a decision to only describe them in the first mention at page 4.

9. Please indicate the significance level, if it is 0.05, then in table 3 two parameters show significant correlation with age (BS /TV and Tb. N).

Response: The significant level was set at 0.05 and it is true that two parameters showed statistically significant correlation with age (BS /TV and Tb. N). Correction has been made accordingly.

10. I suggest performing an independent-samples t-test or its nonparametric version (Mann-Whitney) to determine sex differences in all parameters. Performing a partial correlation (evaluating the control variable) might yield interesting results, please reconsider. I would also run an age correlation with respect to gender.

Response: We agree with this suggestion. However, due to the preliminary nature of this study, we have decided that the rest of non-parametric tests will be performed and reported in the subsequent study of large stratified dataset.

11. in the results section, I would start with the ICC since the other results depend on it.

Response: Thank you for this suggestion and we agree. The tables have been rearranged and ICC results (Table 2) were reported immediately after profile of the study subjects (Table 1). 

12.The conclusion is written too presumptuously given the insignificant correlation analysis or not so strong correlation (considering the small sample size) and the poor ICC. According to the present result, I think that an age range whose average age is closer to the minimum age of the sample could bias the results.

Response: We are wholeheartedly agree with this remark. Therefore, we have amended the conclusion as follows.

The results of the correlation of chronological age with the trabecular microarchitectural parameters of 20 CBCT DICOM images indicate that high-resolution CBCT of the trabecular bone structure may be useful for quantifying differences in mandibular trabecular microarchitectures. Further, this study in future serves as a preliminary analysis to devise a prediction model for depicting the chronological age based on the TBM parameters. A large and stratified dataset on age interval is recommended for future study. 

Reviewer 3 Report (New Reviewer)

The paper entitled "Quantifications of Mandibular Trabecular Bone Microstructure 2 using Cone Beam Computed Tomography for Age Estimation" is very interesting. 

This is a pilot study because the number of cases is small, so I suggest to the authors broadening the discussion about applications in the forensic fields. I think is very important to speak about the use of this method on recent skeletal remains, for example in cases of homicide, and on underage persons of unknown age.

I recommend improving in detail the selection of the cases, specifically the choice of the age range (22-43 age range), and the geographic and ethnic origin.

Author Response

We would like to thank the reviewer for the remarks.

This is a pilot study because the number of cases is small, so I suggest to the authors broadening the discussion about applications in the forensic fields. I think is very important to speak about the use of this method on recent skeletal remains, for example in cases of homicide, and on underage persons of unknown age.

Response: Thank you for this wonderful suggestion. We have added several statements pertinent to the use of the method in forensic human identification in introduction section.

I recommend improving in detail the selection of the cases, specifically the choice of the age range (22-43 age range), and the geographic and ethnic origin.

Response: The case selection has been detailed out according to the suggestion in the results section.

Round 2

Reviewer 1 Report (Previous Reviewer 3)

The manuscript was revised according to suggestions on revision, the conclusion of the manuscript was reasonable and the research had a certain application prospect, I think it basically meets the requirements of Biology. 

Reviewer 2 Report (New Reviewer)

The authors have included all the suggestions I recommended. I propose that the article be approved for publication.

This manuscript is a resubmission of an earlier submission. The following is a list of the peer review reports and author responses from that submission.

Round 1

Reviewer 1 Report

////

Reviewer 2 Report

This study aims to correlate the trabecular bone microstructure(TBM) parameters of the mandible to age and gender. The study has several significant issues although the topic is engaging.

1.  The number of data for statistical analysis is too small which is not enough to derive meaningful results. According to the central limit theorem, at least 30 samples for each age group are typically recommended to represent their population. Thus, it is highly questionable whether the result of this study can represent the correlations between TBM parameters and age/gender.

2. It is hard to believe that the results(i.e correlation) of the study were well calculated because the study omitted TBM parameters calculation method. The calculation for TBM parameters is performed in two ways: direct method(3D) and indirect method(2D). And according to the method, the values of parameters vary. Particularly TBM parameters calculated by indirect method for the region of interest tend to rely on bone volume fraction(BV/TV) because Tb.Th, Tb.Sp, Tb.N are calculated by BV/TV.

3. Another majority issue on background introduction is that the authors only focused on the background of technique development, and did not contain any clinical background which is required by this journal.

4. Overall logical flow of the manuscript should be more smooth.

Reviewer 3 Report

This article aimed to establish the relationship between bone microstructure (TBM) parameters and individual's chronological age and gender. The article had certain application value, however, the content and expression of the article were unreasonable.

1) To explore the correlation between TBM and individual’s information at the statistical level, 20 samples were too small.

2) The figures in this article should be labeled and marked.

3) The CBCT analysis of bone features proposed in this paper was not innovative. At the same time, there have been many research achievements in establishing the relationship between bone features and individual information.